# DciA is an ancestral replicative helicase operator essential for bacterial replication initiation

Pierre Brézellec[1,*], Isabelle Vallet-Gely[2,*], Christophe Possoz[2,*], Sophie Quevillon-Cheruel[2] & Jean-Luc Ferat[1,2]

Delivery of the replicative helicase onto DNA is an essential step in the initiation of replication. In bacteria, DnaC (in *Escherichia coli*) and DnaI (in *Bacillus subtilis*) are representative of the two known mechanisms that assist the replicative helicase at this stage. Here, we establish that these two strategies cannot be regarded as prototypical of the bacterial domain since *dnaC* and *dnaI* (*dna[CI]*) are present in only a few bacterial phyla. We show that *dna[CI]* was domesticated at least seven times through evolution in bacteria and at the expense of one gene, which we rename *dciA* (*dna[CI]* antecedent), suggesting that DciA and Dna[CI] share a common function. We validate this hypothesis by establishing in *Pseudomonas aeruginosa* that DciA possesses the attributes of the replicative helicase-operating proteins associated with replication initiation.

[1] Universite de Versailles Saint-Quentin, 45 Avenue des Etats-Unis, Versailles 78000, France. [2] Institute for Integrative Biology of the Cell (I2BC), CEA, CNRS, Univ. Paris-Sud, Université Paris-Saclay, Gif-sur-Yvette 91198, France. * These authors contributed equally to this work. Correspondence and requests for materials should be addressed to J.-L.F. (email: jean-luc.ferat@i2bc.paris-saclay.fr).

Replicative helicases are essential components of the replication machinery. Once loaded onto DNA, these hexameric ring-shaped enzymes translocate onto single stranded DNA, in the $5' \to 3'$ direction in bacteria and in the $3' \to 5'$ direction in archaea and in eukaryotes, to unwind double stranded DNA in front of the replisome. Loading the replicative helicase onto DNA is an essential step in the initiation of replication and different strategies were selected through evolution to perform this function in the three domains of life. In eukaryotes, replicative helicases are loaded as double-hexamers onto double stranded DNA, assisted by Cdt1 and Cdc6 (ref. 1). In archaea, factors related to Cdt1 and Cdc6 appear to be at work[2]. In bacteria, two loading mechanisms have been described[3]. They are mediated by two different, but related, proteins that assist the delivery of the replicative helicase onto single stranded DNA, Ec_DnaC and Bs_DnaI. In Escherichia coli, the interaction between the helicase and Ec_DnaC reshapes ('breaks') the ring-like structure of the homo-hexameric helicase into an open helicoidal architecture, competent for strand loading[4]. In Bacillus subtilis, two proteins are involved in the delivery of the replicative helicase; Bs_DnaI mediates the assembly of the hexameric ring around single stranded DNA in a mechanism that is stimulated by Bs_DnaB[5].

While DnaI and DnaC are essential for replication initiation in B. subtilis and E. coli, respectively, some organisms appear to lack the gene encoding either of these two proteins. For example, the genome of Pseudomonas aeruginosa, a Gammaproteobacteria closely related to E. coli, contains no gene homologous to dnaC or dnaI. How, then, are replicative helicases managed in this organism? We decided to address the question by drawing the distribution map of dnaC and dnaI (dna[CI]) in the bacterial kingdom. From this study, we establish that the Dna[CI] strategies cannot be regarded as prototypical of the bacterial kingdom anymore since both dnaC and dnaI are present in only 5 of 26 bacterial phyla. We show that dna[CI] was acquired at least seven times through evolution by domestication of distinct but related phage genes at the expense of one gene, which we rename dciA (dna[CI] antecedent). We show that dciA is present in most bacterial phyla (23 out of 26) and systematically lost upon the advent of dna[CI], suggesting that DciA and Dna[CI] share a common function. In agreement with this hypothesis, we show that DciA interacts with its cognate helicase, we establish in P. aeruginosa that DciA is essential for replication initiation and we show that a depletion of the protein results in a blockage of the initiation of replication after the formation of the open complex. The discovery of DciA, a factor unrelated to Dna[CI], opens new perspectives regarding the evolution, mechanisms and function of the management of the replicative helicase during the initiation of replication in bacteria.

## Results

### dna[CI] genes domestication in bacteria through evolution.
In the model organisms E. coli and B. subtilis, DnaC and DnaI are involved in the loading of the bacterial helicase and have long been considered candidates for such a role throughout the bacterial kingdom. Yet, the collection of proteins containing the ATPase domain common and specific to DnaC and DnaI (that is, the Pfam A domain[6], PF01695, here upon referred to as AAA$^{CI}$) among 1,426 complete bacterial proteomes revealed a highly erratic distribution of the genes encoding these proteins within the bacterial kingdom (Supplementary Data 1). Not a single AAA$^{CI}$-containing protein was identified in 646 out of 1,426 proteomes ($\sim$45%), while in others (for example, Aromatoleum aromaticum EbN1) up to 18 different AAA$^{CI}$-containing proteins were identified. We focused our attention on the

Enterobacteriales because this order encompasses both the model organism E. coli, which contains dnaC, and other species lacking AAA$^{CI}$-carrying genes (Fig. 1a). A phylogenetic tree based on the AAA$^{CI}$-containing proteins identified in Enterobacteriales reveals two contrasting distributions (Fig. 1b). One clade, encompassing the DnaC of E. coli, is composed of protein sequences connected by relatively short branches, indicative of intense selection pressure (Fig. 1b, magenta). Their distribution is monophyletic and congruent with that of the 16S rRNA (Fig. 1a) and the genomic context of the genes encoding these proteins is strictly conserved (Supplementary Fig. 1); these genes possess all the attributes of genomic residency. The distribution of the other AAA$^{CI}$-containing proteins present in the Enterobacteriales is notably different. The protein sequences are connected by longer branches, reflecting weak selection pressure (Fig. 1b, black). Furthermore, their distribution is not congruent with that of the species in which they are identified and the genes are systematically included within mobile elements (Fig. 1b; Supplementary Data 1). Taken together, these data reveal, as suggested earlier[7], that dnaC was recently acquired in Enterobacteriales and arose from the domestication of a phage gene.

We detected at least six additional domestication events of a AAA$^{CI}$-carrying gene in bacteria (Fig. 2, magenta): two in Gammaproteobacteria (1) (one in Aeromonadales and another in a restricted clade of Alteromonadales), two in Firmicutes (2) (one in Clostridia and another in Bacilli), one in Aquificae (3) and one in Chloroflexi (4) (Supplementary Figs 1, 2 & 3). The case of the dna[CI] genes found in Tenericutes (5) remains unclear. Although we identified a clade of domesticated dna[CI] genes in Tenericutes, we were unable to establish whether the domestication of dna[CI] in Tenericutes and in Bacilli were distinct events. Overall, 21 out of the 26 bacterial phyla investigated are lacking a resident dna[CI] gene (Fig. 2, cyan and black).

### dciA loss upon advent of dna[CI].
Since proteins that assist replicative helicases during replication initiation were shown to be required in eukaryotes, archaea and bacteria[1,2,5,8], we wondered whether other factors were specified in 'non-dna[CI]' bacteria to assist the replicative helicase in replication initiation. Again, we restricted our investigation to the Enterobacteriales to address this hypothesis, because this order encloses closely related species of both kinds—with or without a resident dna[CI] gene. Our screen, aimed at collecting the protein domains that were systematically and exclusively associated with the Dna[CI]-lacking proteomes (Methods), returned a unique protein domain of unknown function, PF05258 (Fig. 1a). Strikingly, this domain is largely present throughout the whole bacterial kingdom (Fig. 2, cyan). Except for a few species in which neither dna[CI] nor a PF05258-carrying gene was identified (Supplementary Data 1), PF05258-carrying genes are systematically and exclusively present in genomes lacking resident dna[CI] (Fig. 2). Remarkably, the genomic context of PF05258-carrying genes is very well conserved. In 15 out of the 23 phyla in which this gene was identified (Actinobacteria, Fusobacteria, Elusimicrobia, Spirochaetes, Bacteroidetes, Chlorobi, Planctomycetes, Fibrobacteres, Gemmatomonadetes, Chlamydiae, Thermotogae, Thermus-Deinococcus, Dyctioglomi, Firmicutes and Synergistetes), it is located within the replication operon [dnaA-dnaN-recF-gyrB-gyrA] between recF and gyrB or, when the operon is scattered, next to either recF or gyrB. Other conserved, yet minor, genomic contexts were identified in Proteobacteria, suggesting that major DNA rearrangements occurred early in the history of phylum. In beta- and in Gammaproteobacteria, PF05258-carrying genes are located between secA and lpxC, whereas they were identified between mutY and trx-smc and next to uvrC in Alphaproteobacteria and in Epsilonprotebacteria,

respectively. We did not detect conserved genomic location of PF05258-carrying genes in Deltaproteobacteria, Cyanobacteria, Nitrospirae, Verrumicrobia, Thermodesulfobacteria, Deferribacteres and Acidobacteria.

Taken together, their distribution and limited number of genomic locations indicate that PF05258-carrying genes preceded the acquisition of *dna[CI]* in bacteria and that PF05258-carrying genes were lost upon the advent of *dna[CI]*. For these two reasons we renamed the PF05258-carrying gene *dciA* for *dna[CI]* antecedent.

*dciA* genes are almost exclusively genomic—we detected only one instance of 'mobility' of this gene in the genus *Streptomyces* (Supplementary Fig. 4). *dciA* genes specify a highly basic polypeptide, containing occasionally a Zinc finger located in the C-terminal domain of the protein (for example, in Cyanobacteria). DciA, however, contains no AAA$^+$ domain, which was shown to be critical in DnaC for the loading of the replicative helicase[4].

**DciA is essential during replication initiation.** To test whether DciA was similarly critical for the bacterial replicative helicase function, we investigated the involvement of DciA during replication initiation in *P. aeruginosa* (*Pa*), a Gammaproteobacteria. Firstly, we assayed the interaction between DciA and DnaB using a bacterial two-hybrid system. We showed that *Pa*_DciA interacts strongly with the full-length helicase *Pa*_DnaB$^{FL}$ (Fig. 3a). In addition, we established that this interaction is specific, since no interaction was detected between the replicative helicase of *E. coli*, *Ec*_DnaB$^{FL}$, and *Pa*_DciA, despite the high degree of homology between the two helicases (61.4% identity) and their interaction in the two-hybrid system (Fig. 3a). Secondly, we demonstrated that *dciA* is essential. We deleted the *dciA* gene in a strain expressing DciA from a thermosensitive plasmid [p$^{ts}$(*dciA*)] and showed that this strain could not give rise to a viable colony at a non-permissive temperature of 42 °C. Thirdly, we assessed the implication of DciA in replication initiation in *P. aeruginosa* by analyzing *dciA*$^+$/p$^{ts}$(*dciA*) and Δ*dciA*/p$^{ts}$(*dciA*) cells cultivated at 42 °C. Under this non-permissive condition, p$^{ts}$(*dciA*) cannot replicate; the plasmid is passively distributed into new-born cells for a few generations until plasmid-less cells are produced, which occurs after about 4 h of incubation (Fig. 3b). Hence, Δ*dciA*/p$^{ts}$(*dciA*) cells cultivated at 42 °C are a mix of phenotypically wild type (Δ*dciA* cells still containing a single copy of p$^{ts}$(*dciA*)) and mutant cells (plasmid-less Δ*dciA* cells). Two hours after the first p$^{ts}$(*dciA*)-less cells were produced (∼2 generation times), samples were fixed for epifluorescent microscopy and cytometry analyses. DNA staining revealed that the *dciA*$^+$/p$^{ts}$(*dciA*) control cells cycled normally; DNA is homogeneously distributed within the cells (Fig. 3c left). In contrast, Δ*dciA*/p$^{ts}$(*dciA*) cells appear heterogeneous. In some cells, the DNA is asymmetrically distributed within the two daughter cells or chunked by the septum or both, while others appear to have lost their genomic DNA, revealing a defect in the coordination of cell division with replication. The level of fluorescence expected to be associated with plasmid DNA is orders of magnitude lower than that of the genomic DNA (one thousandth of the total fluorescence of the cell), implying that the loss of the plasmid

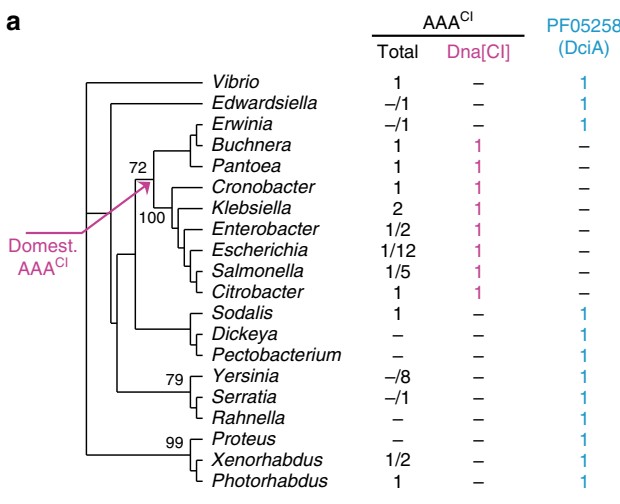

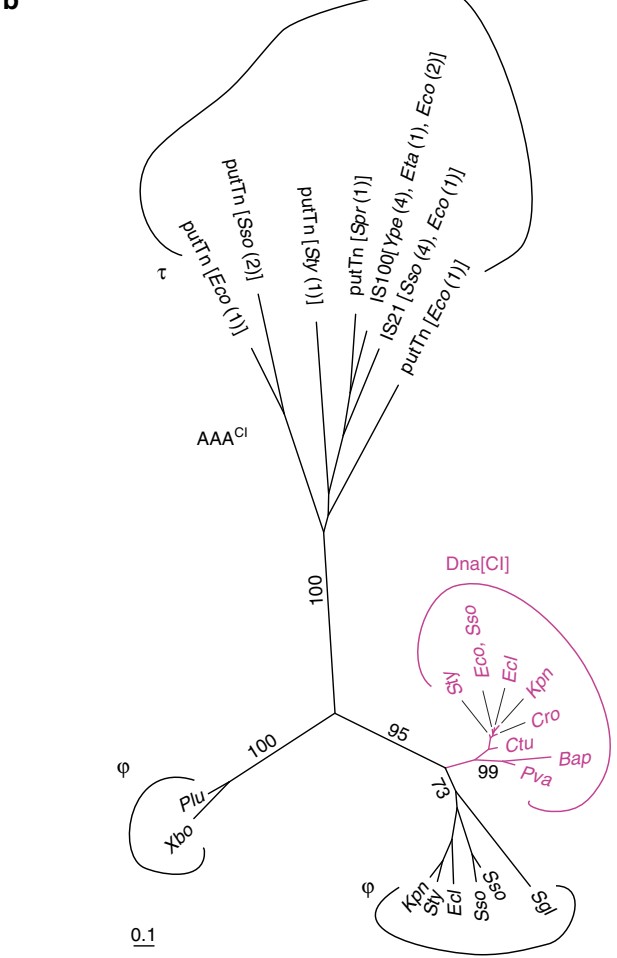

**Figure 1 | *dciA* is systematically and exclusively present in enterobacteriale genomes lacking resident *dna[CI]* genes.**
(**a**) Phylogenetic tree of Enterobacteriales based on DnaA-DnaB-DnaX-DnaE-concatenated protein sequences of each species analysed (Methods). The number of distinct AAA$^{CI}$- and PF05258-containing proteins (total and Dna[CI]) within each genus (minimum/maximum) is indicated. *Vibrio cholerae* was used as an out-group. Dna[CI] and DciA are indicated in magenta and cyan, respectively. The probable origin of Dna[CI] in the Enterobacteriales is pointed at with a magenta arrow. (**b**) Phylogenetic tree of the AAA$^{CI}$-containing proteins found in Enterobacteriales. Strains are listed in the dataset of the Methods section. Dna[CI] proteins are in magenta. Species, in which AAA$^{CI}$-carrying transposons were found, are between square brackets and copy number of each element found in a given genome is between parentheses. τ: transposon, φ: phage. Bootstrap values of interest are provided. Scale bar represents 0.1 substitution per site.

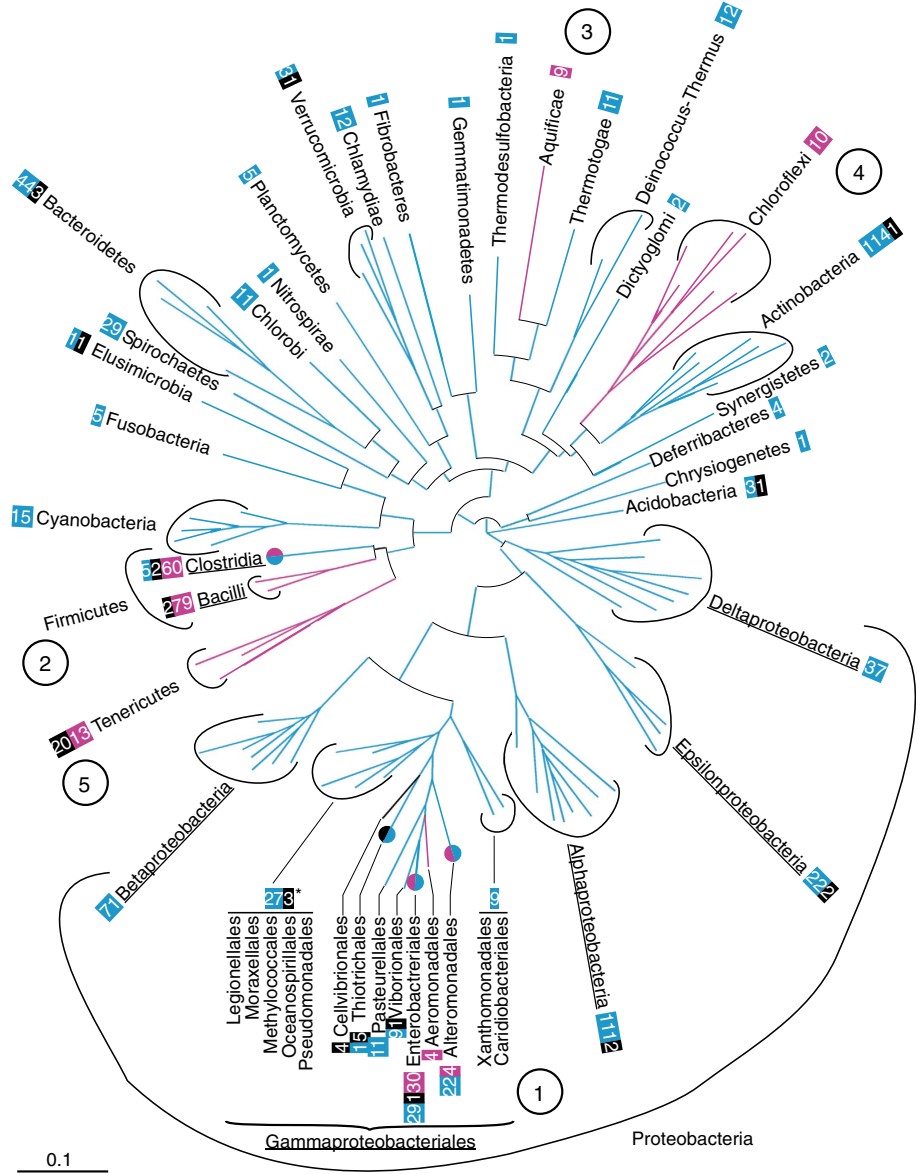

**Figure 2 | Distribution of domesticated *dnaC* and *dciA* in the bacterial domain.** Distribution of the bacterial phyla investigated on a 16S rRNA-based circular phylogenetic tree (Methods). When the presence of *dna[CI]* or *dciA* within a phylum (bold) was not homogeneous, classes (underlined) or orders (normal) are specified. Numbers boxed refer to the number of species' genomes (and not strains) containing domesticated *dna[CI]* (magenta), *dciA* (cyan) or neither (black). Magenta and cyan splitted circles: orders in which *dciA* and *dna[CI]* genes were identified. *The 3 species that are devoid of *dna[CI]* and *dciA* belong to the Oceanospirillales. The order Thiotrichales is identified with a cyan/black circle because most species lack *dciA* and *dna[CI]*. (1) Gammaproteobacteria, (2) Firmicutes, (3) Aquificae, (4) Chloroflexi and (5) Tenericutes. Scale bar represents 0.1 substitution per site.

cannot by itself explain the disappearance of the fluorescence (Fig. 3c right). To circumvent the perturbation created by the apparent absence of coordination between replication and division in DciA-depleted cells, we added Ceftazidime (CAZ), an inhibitor of FtsI in *P. aeruginosa*. Adding CAZ therefore blocks cell division and lead to cell filamentation[9]. Cells were grown at non permissive temperature as previously described, CAZ was then added and growth was prolonged for 2 h under the same conditions before fixation for cytometry analysis (Fig. 3d, Methods). During the incubation with CAZ, DNA content in the control cells ($dciA^+$/p$^{ts}$($dciA$)) rose from a continuous amount of (1 to 2) to (4 to 8) genome-equivalents per cell, indicating that the succession of the replication cycles was not affected by the addition of the drug (follow the abscissa in Fig. 3d, left). In cultures of $\Delta dciA$/p$^{ts}$($dciA$) cells grown at a

non-permissive temperature, two new clusters of cells containing integral numbers of chromosomes ($\alpha$ and $\beta$) appear on the cytogram (boxed in Fig. 3d, right), in addition to the clump of cells that behaved like isogenic $dciA^+$ cells, that is, the cells contain a continuous amount of 4–8 genome equivalents. The two clusters of cells containing integral numbers of chromosomes are composed of elongated cells (indicated here by the increase of the cell mass) over the time of incubation with CAZ and ongoing rounds of replication were completed, while no new round of replication was initiated. This demonstrates that replication initiation is specifically blocked upon depletion of DciA.

We analysed the status of the origin of replication of *P. aeruginosa* in DciA-depleted cells by treating $dciA^+$/p$^{ts}$($dciA$) and $\Delta dciA$/p$^{ts}$($dciA$) cells incubated at a non permissive temperature with potassium permanganate (KMnO$_4$), a chemical

that targets pyrimidines—mostly thymidines—located on single-stranded DNA and modifies them into roadblocks for polymerases[10]. The analysis by primer extension of the *oriC* region[11] revealed strong DNA polymerization stops within the DNA unwinding element (DUE) region (from positions 18–46 downstream of the DnaA box R1) specifically in the Δ*dciA*/p[ts](*dciA*) cells (Fig. 3e). Thus, replication initiation is blocked after the formation of the open complex in DciA-depleted cells, as in DnaC-inactivated *E. coli* cells[12], suggesting that the two proteins participate at the same stage of replication initiation.

## Discussion

Hence, DciA is an ancestral bacterial replicative operator. The gene specifying this protein is widely distributed among bacteria, arguing for a long-standing presence in the bacterial kingdom

and possibly at its root; the conservation of the genomic context of *dciA* (that is, in the replication operon) in most bacterial phyla further supports this hypothesis.

*dciA* and *dna[CI]* specify two types of replicative helicase operators involved in replication initiation. Either one is found in bacteria, revealing that replicative helicase operators are essential and inseparable components of the bacterial replication initiation process. Only a limited number of species (49) are apparently lacking either *dciA* or *dna[CI]*. Most of them contain either a small genome or are host-associated (32 out of 49), suggesting that the lack of *dna[CI]* and *dciA* might have been consequential to the reduction of the size of the genome or to the adaptation of the organisms to a new environment. A few other species lacking both *dna[CI]* and *dciA*, however, appear to be isolated within a homogeneous clade that regroups *dna[CI]* or *dciA*-specifying species. Regarding these species, we cannot exclude that we missed some *dciA* or *dnaC* genes, either because of genome annotation errors or because the signatures of the proteins specified by these genes diverged enough from their consensus domain profiles to prevent their detection. Finally, we identified one order regrouping only *dna[CI]* and *dciA*-lacking organisms, the Cellvibrionales[13]. Whether this genetic isolation is revealing of the acquisition of another replicative helicase operator gene, different from *dciA* and *dna[CI]* is an open question that would be interesting to investigate.

It is interesting to note that except for their distribution in the Enterobacteriales and the Alteromonadales, in which the genetic exaptation of the phagic ancestor of *dna[CI]* is recent (that is, members of these orders fall into Dna[CI]- or DciA-specifying organisms), *dna[CI]* genes, when identified, are present in every species of a complete order-class-phylum; the 5 *dna[CI]*-lacking species classified as Clostridiales in Fig. 2, in which *dciA* was

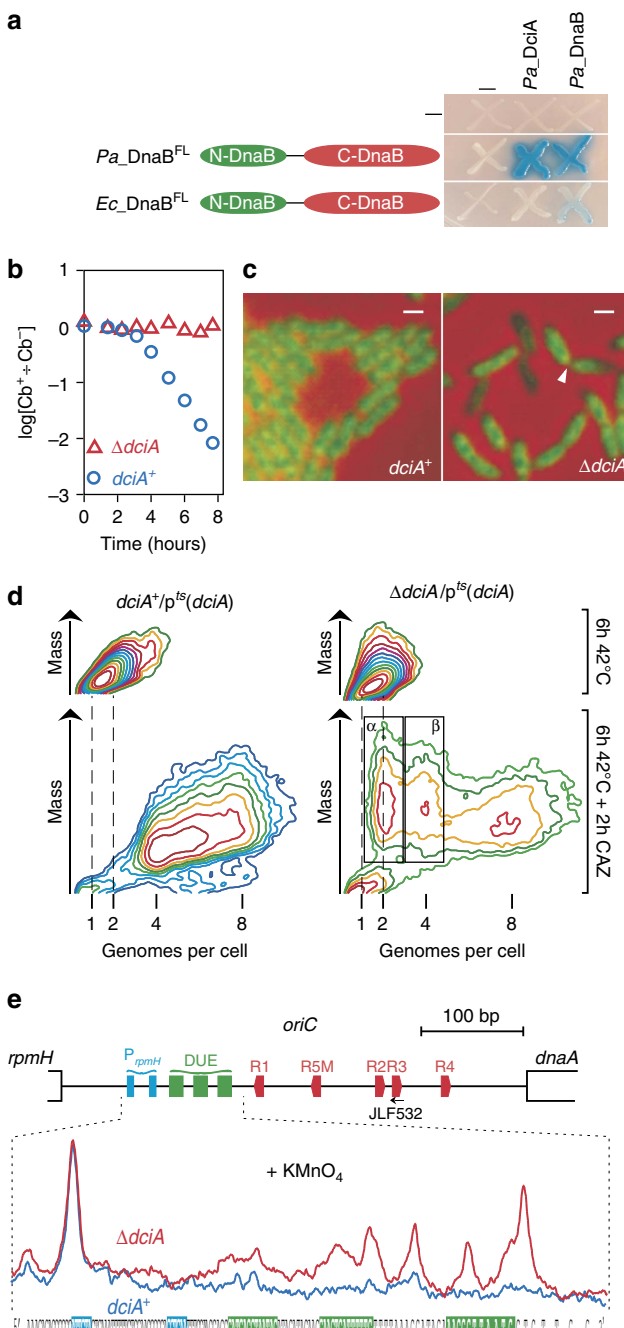

**Figure 3 | DciA is essential during replication initiation in *P. aeruginosa*.** (**a**) Interaction between DnaB and DciA was monitored by the two-hybrid system after 2 days of incubation at 30 °C (Methods). The full length replicative helicase of *P. aeruginosa* (Pa_DnaB[FL]) and *E. coli* (Ec_DnaB[FL]) were assayed against the DciA protein of *P. aeruginosa* (Pa_DciA) or Pa_DnaB[FL] (Methods). (**b**) Cultures of exponentially growing Δ*dciA*/p[ts](*dciA*) and *dciA*[+]/p[ts](*dciA*) cells (30 °C) were transfered at 42 °C. Samples were taken at various time intervals and plated on MMA medium containing (Cb[+]) or not containing (Cb[−]) Carbenicillin (resistance to which is brought by p[ts](*dciA*)) and incubated for 24 h at 30 °C. The ratio of the cfu counted on plates supplemented with or without Carbenicillin (log[Cb[+] ÷ Cb[−]]) is plotted over time of incubation at 42 °C for each culture. (**c**) *dciA*[+]/p[ts](*dciA*) and Δ*dciA*/p[ts](*dciA*) cells grown for 6 h at 42 °C were stained with HOECHST (green) and analysed by epifluorescence (Methods). The white arrowhead points to a septum that pinches DNA at midcell. Scale bar represents 1 μm. (**d**) Contour plot of cytograms of the *dciA*[+]/p(*dciA*) and Δ*dciA*/p(*dciA*) cells before and after 2 h of incubation with CAZ. 'Genomes per cell' reports the amount of DNA per cell relative to the amount of DNA measured in stationary phase cells. Since *Pseudomonas* is a rod-shaped bacterium, mass is roughly proportional to cell size, and therefore indicative of the level of elongation. The clusters of cells in which replication initiation is blocked are boxed. (**e**) Replication initiation is blocked after the open complex formation in DciA-depleted cells. The position relative to the DnaA boxes (red pentagons), the 13mer (green rectangles) and the promoter sequences of *rpmH* (cyan rectangles) between *dnaA* and *rpmH* are indicated. The autoradiography of the primer extension experiment performed on DNA of KMnO4-treated *dciA*[+]/p[ts](*dciA*) (blue) and Δ*dciA*/p[ts](*dciA*) cells (red) was scanned and the profiles obtained were superimposed on the sequence of the origin of replication. All experiments were performed at least 3 times (except for **b**, which was only performed twice) independently, and a representative set of each experiment is shown here.

identified within the replication operon, are indeed excluded phylogenetically from the Firmicute branch. We cannot therefore exclude that upon domestication of a phagic *dna[CI]* gene, significant modifications of the physiology of the cell occurred that forced the isolation of these cells into new species and hence into new orders-classes-phyla.

The fact that none of the 1,426 genomes analysed co-maintained *dciA* and *dna[CI]* remains a striking discovery. The driving force behind the strong counterselection of *dciA* upon the advent of *dna[CI]* is unknown and may result from incompatible activities or functions specified by Dna[CI] and DciA.

## Methods

**Data sets.** A complete list of the genomes used in this study is available in Supplementary Data 1.

The following strains were used to study the distribution of DnaC and DciA within Enterobacterialles: *Bap*: *Buchnera aphidicola* str. APS (Acyrthosiphon pisum), *Cro*: *Citrobacter rodentium* ICC168, *Ctu*: *Cronobacter turicensis* z3032, *Eta*: *Edwardsiella tarda* EIB202, *Ecl*: *Enterobacter cloacae* subsp. dissolvens SDM, *Eco*: *Escherichia coli* O150:H5, *Kpn*: *Klebsiella pneumoniae* subsp. pneumoniae NTUH-K2044, *Pva*: *Pantoea vagans* C9-1, *Plu*: *Photorhabdus luminescens* subsp. laumondii TTO1, *Sty*: *Salmonella typhi*, *Spr*: *Serratia proteamaculans* 568, *Sgl*: *Sodalis glossinidius* str. 'morsitans', *Sso*: *Shigella sonnei* Ss046, *Xbo*: *Xenorhabdus bovienii* SS-2004, *Ype*: *Yersinia pseudotuberculosis* serotype IB.

The following strains were used to identify *dciA* in Enterobacterales. In-group set: *Proteus mirabilis* HI4320, *Pectobacterium carotovorum* subsp carotovorum PC1, *Photorhabdus luminescens* subsp laumondii TT01, *Dickeya dadantii* Ech703, *Yersinia pestis* 632, *Pectobacterium wasabiae* WPP163, *Serratia proteamaculans* 568, *Yersinia enterocolitica* serotype O8 biotype 1B 8081, *Dickeya dadantii* 3937, *Erwinia chrysanthemi* 3937, *Photorhabdus asymbiotica* subsp asymbiotica ATCC 43949, *Xenorhabdus luminescens* 2, Dickeya zeae Ech1591. Out-group set: *Citrobacter rodentium* ICC168, *Citrobacter freundii* biotype 4280, *Enterobacter cloacae* SCF1, *Cronobacter turicensis* z3032, *Escherichia coli* K12, *Salmonella typhimurium* 90371, *Klebsiella pneumoniae* subsp pneumoniae MGH 78578.

The following strains were used as representative of bacterial phyla-classes-orders to build up the phylogenetic tree presented in Figure 2: Actinobacteria: *Acidimicrobium ferrooxidans* DSM 10331, *Mycobacterium bovis* BCG str. Pasteur 1173P2, *Bifidobacterium longum* subsp. infantis ATCC 15697, *Coriobacterium glomerans* PW2, *Rubrobacter xylanophilus* DSM 9941, *Conexibacter woesei* DSM 14684; Aquificae: *Aquifex aeolicus* VF5; Bacteroidetes: *Bacteroides fragilis* CR626927, *Flavobacterium johnsoniae* UW101, *Rhodothermus marinus* DSM 4252, *Cytophaga hutchinsonii* ATCC 33406; Chlamydiae: *Chlamydia trachomatis* L1/440/LN; Chlorobi: *Chlorobaculum tepidum* TLS; Chloroflexi: *Anaerolinea thermophila* UNI-1, *Chloroflexus aurantiacus* Y-400-fl, *Herpetosiphon aurantiacus* CP000875, *Thermomicrobium roseum* DSM 5159, *Sphaerobacter thermophilus* DSM 20745, *Dehalococcoides ethenogenes* strain 195; Chrysiogenetes: *Desulfurispirillum indicum* S5; Deferribacteres: *Deferribacter desulfuricans* SSM1; Thermus-Deionococcus: *Deinococcus deserti* VCD115, *Thermus thermophilus* HB27; Dictyoglomi: *Dictyoglomus thermophilum* DSM 3960; Elusimicrobia: *Elusimicrobium minutum* Pei191; Fibrobacteres: *Fibrobacter succinogenes* subsp. succinogenes S85; Fusobacteria: *Fusobacterium nucleatum* subsp. nucleatum; ATCC 25586; Gemmatimonadetes: *Gemmatimonas aurantiaca* T-27; Nitrospirae: *Thermodesulfovibrio yellowstonii* YP87; Planctomycetes: *Rubinisphaera brasiliensis* DSM 5305; Alphaproteobacteria: *Caulobacter vibrioides* CB15, *Parvularcula bermudensis* HTCC2503, *Rhizobium radiobacter* C58, *Rhodobacter sphaeroides* ATCC 17029, *Polymorphum gilvum* SL003B-26A1, *Rhodospirillum rubrum* ATCC 11170, *Rickettsia rickettsii* str. 'Sheila Smith', *Sphingomonas wittichii* RW1; Betaproteobacteria: *Burkholderia mallei* ATCC 23344, *Thiobacillus denitrificans* ATCC 25259, *Methylotenera mobilis* JLW8, *Neisseria gonorrhoeae* NCCP11945, *Nitrosomonas europaea* ATCC 19718, *Azoarcus* sp. BH72, *Gallionella capsiferriformans* ES-2; Deltaproteobacteria: *Bdellovibrio bacteriovorus* HD100, *Desulfarculus baarsii* DSM 2075, *Desulfobulbus propionicus* DSM 2032, *Desulfovibrio vulgaris* str. 'Miyazaki F', *Geobacter sulfurreducens* subsp. null KN400, *Myxococcus xanthus* DK 1622, *Syntrophobacter fumaroxidans* MPOB; Epsilonproteobacteria: *Helicobacter pylori* 26695-1CH, *Sulfurovum* sp. NBC37-1, *Nautilia profundicola* AmH, *Nitratifractor salsuginis* DSM 16511; Gammaproteobacteria: *Acidithiobacillus ferrooxidans* ATCC 53993, *Aeromonas hydrophila* subsp. hydrophila ATCC 7966, *Alteromonas macleodii* str. 'Deep ecotype', *Saccharophagus degradans* 2-40, *Dichelobacter nodosus* VCS1703A, *Nitrosococcus oceani* ATCC19707, *Escherichia coli* str. K-12 substr. W3110, *Legionella pneumophila* str. Paris, *Methylococcus capsulatus* ACM 3302, *Halomonas elongata* DSM 2581, *Pasteurella multocida* subsp. multocida str. Pm70, *Psychrobacter arcticus* 273-4, *Pseudomonas aeruginosa* PAO1, *Thioalkalimicrobium cyclicum* ALM1, *Vibrio cholerae* O1 biovar eltor str. N16961, *Xylella fastidiosa* subsp. null 9a5c; Spirochaetes: *Treponema pallidum* subsp. pallidum str. Nichols; Synergistetes: *Thermanaerovibrio acidaminovorans* DSM 6589; Tenericutes: *Acholeplasma laidlawii* PG-8A, *Mesoplasma florum* L1 ATCC 33453, *Mycoplasma bovis* PG45; Thermodesulfobacteria: *Thermodesulfobacterium* sp. OPB45;

Thermotogae: *Thermotoga maritima* MSB8; Verrucomicrobia: *Opitutus terrae* PB90-1, *Coraliomargarita akajimensis* DSM 45221, *Akkermansia muciniphila* ATCC BAA-835; Acidobacteria: *Granulicella tundricola* MP5ACTX9; Bacilli: *Bacillus subtilis* 168, *Lactobacillus casei* ATCC 334; Clostridia: *Clostridium tetani* E88 Massachusetts; Cyanobacteria: *Nostoc* sp. PCC 7120, *Prochlorococcus marinus* subsp. null str. AS9601, *Gloeobacter violaceus* PCC 7421, *Trichodesmium erythraeum* IMS101, *Synechocystis* sp. PCC 6803.

**Phylogenetic analyses.** An alignment of the protein (a crude alignment generated using the program ClustalW and refined by hand) or nucleotide sequences (extracted from the RDP[14]) was fueled into PhyLM (v. 3.0) and 100 bootstrap replicates were generated for each analysis[15]. A consensus tree was eventually obtained by running the program CONSENSE and fed as an input tree into PhyML. Significant bootstrap scores (arbitrarily above 70%) or those associated with branches of special interest (for example, those bootstrap scores associated with branches of domesticated *dnaC*) are indicated.

**Identification of *dciA*.** We assembled a set of 'in-group' organisms, that is, Enterobacteriales genomes that do not contain a copy of the domesticated *dnaC* gene, and a set of 'out-group' organisms, that is, Enterobacteriales genomes that contain a domesticated *dnaC* gene. Then, we designed a code, divided in three steps, to perform the genomic screen (Supplementary Software):

(1) Associate to each organism belonging to the in-group its own set of Pfam-A domains. Then, intersect all the computed sets and collect the intersected domains in one set (in-group domain set).
(2) Associate to each organism of the out-group its own set of Pfam-A domains. Then, regroup all domains in one set (out-group domain set).
(3) Keep and display the domains that belong solely to the in-group organism, that is, remove from the in-group domain set each domain that appears also within the out-group domain set.

**Plasmids and strains.** *E. coli* DH5alpha (Invitrogen) was used as the recipient strain for all plasmid constructions, whereas *E. coli* strain β2163 (ref. 16) was used to transfer plasmids into *P. aeruginosa*.

The thermosensitive plasmid pREP637 was generated by cloning the PCR-amplified thermosensitive origin of replication of plasmid pSS255 (ref. 17) (primers 5′ AAAGGC**GCTCTTC**GAAGCGGTGGCCACGGCCTCTAG3′ and 5′ TTAGCC**AACGTT**GACGCCAAGGGTGAATC 3′) on a *Sap* I—*Acl* I fragment into the pVDIV plasmid[18]. Next, the *bla* gene was amplified by PCR (primers 5′ AT**GGATCC**ATGTGCGCGGAACCCCTA 3′ and 5′ AT**GGATCC**TATGAG TAAACTTGGTCT 3′) and cloned into the *Bgl* II site of the *aacC1* gene resulting in plasmid pREP639. The *dciA* gene and its promoter (from the position + 4937672 on the genome of PAO1 to the TAA encoding the stop codon) were amplified by PCR and cloned onto a *Hind* III—*Not* I fragment into pREP639, resulting in p*ts*(*dciA*) (*dciA* was incorrectly annotated in PAO1; the genuine ATG codon starts at position + 4937754). Plasmid and oligos sequences are available upon request.

The deletion construct for the *dciA* gene (PA4405) was generated by amplifying flanking regions by PCR (flanking region 1: 5′ ATAT**TCTAGA**TAGGTCGAC TGGCTGATG 3′ and 5′ ATAT**GAATTC**CATGGCGGGAAGCCTGGG 3′; flanking region 2: 5′ ATAT**GAATTC**AAAAGCCGGCAAGAATAA 3′ and 5′ AT**GGTACC**CGAAGTCAGCGTGGAAGA 3′) and then splicing the flanking regions together by overlap extension PCR; deletion was in-frame and contained the 6-bp linker sequence 5′-GAATTC-3′. The resulting PCR products was cloned as *Xba* I—*Hind* III fragments into a non-replicative plasmid containing the *sacB* gene, pEXG2 (ref. 19), yielding plasmid pEXM*dciA*. *P. aeruginosa* was mated with β2163 containing the resulting plasmid, and PAO1::pEXM*dciA* integrants were selected on LB plates supplemented with gentamicin. A replicative thermosensitive plasmid carrying *dciA* was then introduced by electroporation into PAO1::pEXM*dciA*. Transformants were selected at 30 °C on LB plates supplemented with carbenicillin and streaked on LB plates containing 5% sucrose to counterselect cells that contained *sacB* and therefore the integrated plasmid, creating the strains Δ*dciA*/p*ts*(*dciA*) by allelic exchange. Deletion *dciA* was confirmed by PCR (primers 5′ GCGAGGAGATGTAGCCCT 3′ and 5′ GTTCGT ATTCCTGATCCA 3′).

***In vivo* ssDNA assay.** Fresh overnight cultures of cells placed in MMA[20] at 30 °C were diluted in the same medium and grown at 42 °C for 6 h. Extemporaneously made potassium permanganate (0.3 M boiled for 5 min) was added for 1 min to cultures (25 to 50 ml) at a final concentration of 3 mM. The permanganate mixture was neutralized by adding an equal volume of NS (100 mM NaCl, 10 mM Tris-HCl (pH 8.0), 1 mM EDTA (pH 8.0) and 5 mM DTT) and chilled on ice. Cells were pelleted at 4 °C for 15 min, washed twice with 1 ml of 50 mM Tris-HCl (pH 8.0) and resuspended in TE (10 mM Tris-HCl (pH 8.0), 1 mM EDTA (pH 8.0)).

Thirty microlitre SDS (10%) and 3 μl freshly prepared Proteinase K (20 mg ml$^{-1}$) were added to 567 μl of resuspended cells. The lysis mixture was incubated for an hour at 37 °C, then complemented with 100 μl of NaCl (5 M) and 80 μl of Hexadecyl Trimethylammonium Bromide (CTAB), and further incubated for 10 min at 50 °C. Cell debris was discarded by two chloroform extractions. DNA

was then extracted twice with phenol-iso-amyl alcohol (24:1), precipitated, washed twice in ethanol (70%), air dried, and resuspended in water.

Primer extension is adapted from Park *et al.*[21] Linear amplification (30 cycles) of genomic DNA was performed with 5′ ($^{32}$P) radiolabelled primer (*oriC* senJLF532 5′ GAAAAAACCGCTGTGGATAAC 3′) in PCR standard conditions.

**Microscopy.** Cells were grown until an OD$_{550}$ of 0.1, and fixed with an equal volume of a 1× PBS solution containing 5% paraformaldehyde and 0.06% glutaraldehyde. After an overnight incubation at 4 °C, the cells were washed twice in PBS and then incubated in a solution of 1 μg ml$^{-1}$ HOECHST 33258 (Thermofisher). After 20 min incubation at room temperature, the cells were washed in 1× PBS, spread out on agarose pads and observed immediately using a Leica DM6000 microscope, a coolsnap HQ CCD camera (Roper) and Metamorph software.

**Flow cytometry.** The procedure was extensively described previously[8]. Briefly, cells to be analysed were fixed by adding 5 volumes of 70% ethanol per volume of sample. Before analysis, the cells were washed twice in filtered TE (Tris-HCl (pH 7.5) 10 mM, EDTA (pH 8) 1 mM]), resuspended in TE supplemented with RNase A and propidium iodide (10 μg ml$^{-1}$, each), incubated for 2 h at 37 °C and analysed. Stained cells were analysed on a PARTEC Particle Analyzing System, PAS III. 100,000 cells were counted per run and data were analysed with the Flowmax software, version 2.52, and WinMDI 2.9.

**Bacterial two-hybrid system.** We used the BATCH system—based on the reconstitution of adenylate cyclase activity—to detect *in vivo* interactions between proteins[22]. The following sequences encoding polypeptides to be analysed by the two-hybrid system were amplified by PCR and cloned into *Xba* I—*Kpn* I pre-digested plasmids pKT25 (pKNT25) and pUT18 (pUT18C) to produce in-frame fusions with the T25 and the T18 subunit coding sequences, respectively: *Ec*_DnaB$^{FL}$ (positions 2-471, primers: 5′ TCCTTC**TCTAGA**GGCAGGAAATAA ACCCTTC 3′ and 5′ TCCTTC**GGTACC**TCGTCGTCGTACTGC 3′), *Pa*_DnaB$^{FL}$ (positions 2-464, primers: 5′ CTTGCGTCGA**TCTAGA**GAACGAGATCAC 3′ and 5′ CAAGCCG**GGTACC**TCGTCCTCGAAG 3′) and *Pa*_DciA (positions 21-153, primers: 5′ GCCGCT**TCTAGA**GCTGTTCGCCGAAGCCCAG 3′ and 5′ ATTTC GGG**GGTACC**TCTTGCCGGCTTTTC 3′). The DHM1 strain was transformed with different combinations of plasmids (pKT + pUT derivatives) and plated on MMA medium complemented with Maltose, Ampicilin, Kanamycin, IPTG and X-GAL as required[22]. The plates were analysed after an incubation of 2 days at 30 °C.

**Data availability.** Accession numbers for the analysed genomes are provided in Supplementary Data 1. All data, primers, plasmids and strains are available upon request from the authors.

The code designed for the genomic analysis (described in the 'Identification of *dciA*' paragraph) is available upon request.

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

## Acknowledgements

We are indebted to Jim Henle for critical reading and improvement of the manuscript. We are thankful to François-Xavier Barre, Frédéric Boccard, Rita Cha, Kathrin Marheineke, François Michel, Herman van Tilbeurgh and Bénédicte Michel for critical discussions and helpful comments. We also thank Sang-Jin Suh (Auburn University, USA) for providing plasmids. This work was supported by the French National Research Council (CNRS).

## Author contributions

P.B. was involved in the design of the study, performed the *in silico* experiments, analysed data. I.V.-G. was involved in the design of the study, participated to the construction of the *P. aeruginosa* strains, performed the microscopic analysis, analysed data. C.P. was involved in the design of the study, constructed plasmids and performed the two-hybrid experiments, analysed data. S.C. was involved in the design of the study, J.-L.F. was involved in the design of the study, participated to the construction of the *P. aeruginosa* strains, performed the *in vivo* ssDNA assay, performed the flow cytometry, analysed data and wrote the paper. P.B., I.V.-G. and C.P. contributed equally to this work. All authors discussed the results and commented on the manuscript.

## Additional information

**How to cite this article**: Brézellec, P. *et al.* DciA is an ancestral replicative helicase operator essential for bacterial replication initiation. *Nat. Commun.* **7,** 13271 doi: 10.1038/ncomms13271 (2016).

