## [Peer review file · Nature Communications]

Reviewers' comments:

Reviewer #1 (Remarks to the Author):

This manuscript describes the identification of a new protein involved in replication initiation in bacteria, named DciA. This is an interesting and novel discovery since most previous work on replication initiation has been performed in the model bacterial organisms *E. coli* and *B. subtilis*, which uses DnaC/I. The approach taken to identify dciA was to search for genes uniquely present in species that do not contain dnaC/I. The identification of a gene involved in replication initiation was indirectly supported by its location inside an operon that also contained other genes involved in replication such as dnaA, dnaN and gyrAB. Importantly, its role in replication initiation was confirmed by experimental studies in *Pseudomonas aeruginosa*.

The manuscript contains quite a few phylogenetic inferences. The tree topologies are important since they are used to infer acquisitions of dnaC/I. Bootstrap support values should be included in all trees to show the statistical support for the diversification patterns, and thereby the reliability of the tree topology. The phylogenetic inferences could be further improved by maximum likelihood and Bayesian analyses. In Figure 1A, Buchnera is shown as an outgroup to the other strains. This is most likely a long branch artifact due to its high AT content and rapid sequence divergence since previous phylogenetic studies have shown that Buchnera clusters closely with *E. coli* and *Salmonella*. Such a placement fits well with the identification of dnaC/I in both Buchnera and *E. coli*. It would be interesting to know in which species neither dnaC/I or dciA have been identified. Do these species have atypically small genomes or do they contain other abnormalities?

Reviewer #2 (Remarks to the Author):

This paper proposes that a group of replicative helicases found in many bacteria of disparate phyla are related to a gene carried by a phage (dciA). The paper presents evidence for independent acquisition of this phage-originated gene during evolution and shows that it can function as a replicative helicase in *Pseudomonas aeruginosa*.

The paper reports interesting evolutionary relationships among distantly related groups of bacteria and proposes an interesting hypothesis for the convergent evolution of an important catalytic pathway involving horizontal gene transfer. The reported results regarding the potential gene transfer of dciA in bacterial belonging to various clades and the supporting observations regarding the uneven distribution of AAA proteins in bacteria are notable. Biochemical studies establish an *in vitro* interaction between dciA and the DnaB and show that dciA is essential for bacterial viability. Potassium permanganate primer extension analyses demonstrate replication elongation defects in dciA mutants.

Although the results are interesting, it is unclear how the evolutionary relationships reported in the paper affect the current paradigm describing a role for replicative helicases during DNA replication. The paper will also be strengthened by including direct biochemical evidence for a role of dciA in the replication initiation process and by a detailed analysis of the relationships among the various forms of dciA in the diverse bacterial phyla. The paper should include measures of reproducibility. For example, how many times were the permanganate primer extension analyses performed and how reproducible were the data? Did independently generated mutants exhibit a consistent phenotype?

Minor

Abstract - delete the number (2) from the third sentence.

Consider including some of the observations reported as "extended data" in the main paper.

Reviewer #3 (Remarks to the Author):

P. Brézellec, I. Vallet-Gely, C. Possoz, S. Quevillon-Cheruel, and H. Ferat: "identification of DciA, an ancestral protein essential to bacterial replication initiation" submitted for publication in Nature Communications.

In this work, the authors note that a large fraction of bacterial species do not appear to encode DnaC or DnaI proteins, which are considered to be necessary for loading the replicative DNA helicase (DnaB) and hence permitting DNA replication initiation. A database search revealed the presence of a novel protein (termed DciA) that is unique to these organisms and which likely fulfills the DnaC/I function. In fact, it is argued that DciA actually represents the ancestral initiation protein, while the DnaC/I proteins are likely transplants for phage genomes (domestication). The authors then go on to demonstrate experimentally in various ways that the DciA protein from a *Pseudomonas* species indeed fulfills the function of interacting with the replicative helicase and facilitating its loading.

Overall, this is very nice and novel work. The arguments that DciA is indeed an ancestral replication protein are highly convincing. It is also revealing that DciA appears to be the predominant protein with this particular function within the bacterial domain.

I have only the following, minor comments for improvement.

1. On page 5, the description of the results of Fig. 3c should be improved. While $\Delta(dciA)$ cells upon plasmid loss at 42°C are expected to become DciA defective and, hence, will lack the ability to initiate replication, it is not immediately clear why they should stain dark (Fig. 3C, right panel). Have they lost their DNA? Some better explanation is needed than just "the detection limit". One might also include a statement whether or not the plasmid contributes in any way to the fluorescence.
2. The flow issue of Fig. 3d could also use a better narrative. The authors state that "two clusters of elongated cells containing integral numbers of chromosomes (2 or 4) are clearly visible on the cytogram" (page 6, lines 5 and 6). I would certainly drop the word "clearly" (it is not so clear, at least to me). A more careful explanation of what the reader is supposed to see in the Fig. 3d would be helpful. Perhaps, some boxes around the presumed "clusters" would be helpful. Also, one should clarify where and how one can see that cells are "elongated", and why they would be elongated.
3. The word HOECHST is misspelled twice: page 12, line 5 from bottom, and page 20, line 2.

Responses to reviewer #1

Reviewer #1 (Remarks to the Author):

The manuscript contains quite a few phylogenetic inferences. The tree topologies are important since they are used to infer acquisitions of *dnaC/l*. Bootstrap support values should be included in all trees to show the statistical support for the diversification patterns, and thereby the reliability of the tree topology.

Authors: We included bootstrap values on every tree. For clarity, however, we provided bootstraps either significant value (above 70 %) or associated with branches of special interest (e.g., the branch from which emerges a clade of domesticated *dnaC*). We wish to maintain Figure 2 as it is, since bootstrap values would apply to the actual species that was used to construct the tree, and not to the phylum to which it belongs and that is solely represented. We thought of representing Figure 2 as a table, but we considered that a “generic” phylogenetic tree (in which orders/classes/phyla were symbolically represented by single species) would have a more visual effect.

The phylogenetic inferences could be further improved by maximum likelihood and Bayesian analyses.

Authors: We redrew all trees (except that of Figure 2) using a ML algorithm (PhyML v. 3.1).

In Figure 1A, *Buchnera* is shown as an outgroup to the other strains. This is most likely a long branch artifact due to its high AT content and rapid sequence divergence since previous phylogenetic studies have shown that *Buchnera* clusters closely with *E. coli* and *Salmonella*. Such a placement fits well with the identification of *dnaC/l* in both *Buchnera* and *E. coli*.

Authors: We agree with the remark of the reviewer. In the previous version, we indicated that *Vibrio cholerae* was used as an out-group in the legend of figure 1. We, however, did not display the complete branch corresponding to *V. cholerae*. As a result, the figure was misleading. We corrected the figure accordingly. To clarify the point raised by the referee, we changed the set of sequences to be compared to establish the phylogenetic tree displayed in Figure 1a. To circumvent the AT bias associated with symbiont genomes pointed by the referee, we analyzed an assembled cluster of essential replication protein sequences (DnaA-DnaB-DnaE-DnaX) instead of 16S rRNA sequences. We changed the materials and methods section accordingly.

It would be interesting to know in which species neither *dnaC/l* or *dciA* have been identified. Do these species have atypically small genomes or do they contain other abnormalities?

Authors: We added pieces of information to the supplementary table that might be relevant to characterize the organisms in which neither *dnaC/l* nor *dciA* was found (genome size, aerobically vs. anaerobically, growth temperature, environment). Altogether, we could not identify a correlation between either of these criteria and the lack of *dnaC/l* and *dciA*, although small genomes seem to be over-represented, as anticipated by the referee.

Responses to reviewer #2

Reviewer #2 (Remarks to the Author):

This paper proposes that a group of replicative helicases found in many bacteria of disparate phyla are related to a gene carried by a phage (*dciA*). The paper presents evidence for independent acquisition of this phage-originated gene during evolution and shows that it can function as a replicative helicase in *Pseudomonas aeruginosa*.

The paper reports interesting evolutionary relationships among distantly related groups of bacteria and proposes an interesting hypothesis for the convergent evolution of an important catalytic pathway involving horizontal gene transfer. The reported results regarding the potential gene transfer of *dciA* in bacterial belonging to various clades and the supporting observations regarding the uneven distribution of AAA proteins in bacteria are notable. Biochemical studies establish an *in vitro* interaction between *dciA* and the DnaB and show that *dciA* is essential for bacterial viability. Potassium permanganate primer extension analyses demonstrate replication elongation defects in *dciA* mutants.

Although the results are interesting, it is unclear how the evolutionary relationships reported in the paper affect the current paradigm describing a role for replicative helicases during DNA replication.

Authors: The results of this paper do not affect the paradigm about a role for replicative helicases during DNA replication. It indicates that the mechanisms of delivery of the replicative helicase onto the DNA that were previously described cannot be seen as prototypical of the bacterial domain anymore. Indeed, the DnaC (in *Escherichia coli*) and DnaI (in *Bacillus subtilis*) proteins are not widespread in the bacterial domain. We identified another protein, which we called DciA, that might be the ancestral replicative helicase operator. We rephrased the abstract and introduction to make this statement clearer.

The paper will also be strengthened by including direct biochemical evidence for a role of *dciA* in the replication initiation process and by a detailed analysis of the relationships among the various forms of *dciA* in the diverse bacterial phyla.

Authors: Cytometry analyses of DciA-depleted cells, as well as primer extension analyses performed on permanganate-modified DNA demonstrate a direct implication of DciA in the replication initiation process. We agree with the referee that additional biochemistry experiments will be necessary to further characterize the molecular mechanisms at play. These experiments, however, require setting up an *in vitro* replication initiation assay. While the production/purification of the factors (DnaB, DciA, DnaA, etc.) involved in replication initiation is on its way, the complete set of proteins is still not available at the moment. This work will be the subject for another paper focused on the mechanistic of the DciA-associated management of replicative helicases.

The paper should include measures of reproducibility. For example, how many times were the permanganate primer extension analyses performed and how reproducible were the data? Did independently generated mutants exhibit a consistent phenotype?

Authors: A statement about measure of reproducibility is now included in the Figure 3 legend ("Experiments were performed at least 5 times independently, and a representative set of experiments is shown here"). Independently generated mutants indeed exhibited the same phenotype.

Minor

Abstract - delete the number (2) from the third sentence.

Authors: The text and bibliography have been reformatted according to Nature Communications requirements.

Consider including some of the observations reported as "extended data" in the main paper.

Authors: The observations regarding the genomic context of PF05258-carrying genes are now fully described in the results section of the paper.

Responses to reviewer #3

Reviewer #3 (Remarks to the Author):

P. Brézellec, I. Vallet-Gely, C. Possoz, S. Quevillon-Cheruel, and H. Ferat: "identification of DciA, an ancestral protein essential to bacterial replication initiation" submitted for publication in Nature Communications.

In this work, the authors note that a large fraction of bacterial species do not appear to encode DnaC or DnaI proteins, which are considered to be necessary for loading the replicative DNA helicase (DnaB) and hence permitting DNA replication initiation. A database search revealed the presence of a novel protein (termed DciA) that is unique to these organisms and which likely fulfills the DnaC/I function. In fact, it is argued that DciA actually represents the ancestral initiation protein, while the DnaC/I proteins are likely transplants for phage genomes (domestication). The authors then go on to demonstrate experimentally in various ways that the DciA protein from a *Pseudomonas* species indeed fulfills the function of interacting with the replicative helicase and facilitating its loading. Overall, this is very nice and novel work. The arguments that DciA is indeed an ancestral replication protein are highly convincing. It is also revealing that DciA appears to be the predominant protein with this particular function within the bacterial domain. I have only the following, minor comments for improvement.

1. On page 5, the description of the results of Fig. 3c should be improved. While $\Delta(dciA)$ cells upon plasmid loss at 42°C are expected to become DciA defective and, hence, will lack the ability to initiate replication, it is not immediately clear why they should stain dark (Fig. 3C, right panel). Have they lost their DNA? Some better explanation is needed than just "the detection limit". One might also include a statement whether or not the plasmid contributes in any way to the fluorescence.

Authors: The description of the figure 3c has been modified to improve clarity. The sentence: *"In contrast, in some $\Delta dciA/p^{ts}(dciA)$ cells, the fluorescence was below detection level, in others, the DNA appeared to be asymmetrically distributed within the two daughter cells or chunked by the septum or both (Figure 3c right), revealing a defect in the coordination between the different steps of the cell cycle."* was replaced by: *"In contrast, $\Delta dciA/p^{ts}(dciA)$ cells appear heterogeneous. In some cells, the DNA is asymmetrically distributed within the two daughter cells or chunked by the septum or both, while others appear to have lost their genomic DNA, revealing a defect in the coordination of cell division with replication. The level of fluorescence expected to be associated with plasmid DNA is orders of magnitude lower than that of the genomic DNA (1/1000 of the total fluorescence of the cell); the plasmid loss cannot by itself explain the disappearance of the fluorescence (Figure 3c right)."*

Yet, it is true that we do not have an explanation as to why anucleated cells are produced upon DciA depletion. Our statement that this observation reveals the conjunction between a lack of replication and a defect in the coordination between replication and division is likely accurate, though. The inhibition of cell division upon addition of CAZ allows the observation of cells blocked for replication initiation by flow cytometry (see below for better narrative of the section). Further investigations will be needed to understand the implication of the lack of DciA in the perturbation of the coordination between replication and cell division.

2. The flow issue of Fig. 3d could also use a better narrative. The authors state that 'two clusters of elongated cells containing integral numbers of chromosomes (2 or 4) are clearly visible on the cytogram' (page 6, lines 5 and 6). I would certainly drop the word "clearly" (it is not so clear, at least to me). A more careful explanation of what the reader is supposed to see in the Fig. 3d would be helpful. Perhaps, some boxes around the presumed "clusters" would be helpful. Also, one should clarify where and how one can see that cells are "elongated", and why they would be elongated.

Authors: The word clearly was dropped and boxes were drawn around the cell clusters. Indications that cell mass (ordinate of the cytograms) measured by flow cytometry is in fact representative of cell length in rod-shaped cells is now provided in the legend of Figure 3. The target of CAZ, its mechanism of action and the filamentation that results from its use (confirmed by a systematic microscopy of flow cytometry analyzed cells) were also added in the main text.

3. The word HOECHST is misspelled twice: page12, line 5 from bottom, and page 20, line 2.

Authors: Spelling of the word Hoechst has been corrected.

REVIEWERS' COMMENTS:

Reviewer #1 (Remarks to the Author):

The points raised in the previous round of reviews have been satisfactorily addressed in this version of the manuscript.

Reviewer #2 (Remarks to the Author):

The revision had addressed most of my concerns. Specifically: 1/ The new abstract is indeed clearer and indeed clarifies the novelty of the findings. 2/ I agree that demonstration of a role in initiation is beyond the scope of the current report and could form the basis of a separate paper. 3/ The number of times experiments were performed are listed.

Minor issues:

1/ In the abstract, perhaps the authors would consider replacing the word "iconic" with "widely recognized".

2/ The authors mention in their letter that the experiments reported in Figure 3 were performed at least 5 times, but the statement in the paper states that the number of repetition was lower (repeated 3 times with the exception of one experiment that was performed twice). Both statement are acceptable, but is the statement in Figure 3 mistaken?

Reviewer #3 (Remarks to the Author):

All critical comments have been addressed satisfactorily.

Response to reviewer's comments

Reviewer #1 (Remarks to the Author):

The points raised in the previous round of reviews have been satisfactorily addressed in this version of the manuscript.

Reviewer #2 (Remarks to the Author):

The revision had addressed most of my concerns. Specifically: 1/ The new abstract is indeed clearer and indeed clarifies the novelty of the findings. 2/ I agree that demonstration of a role in initiation is beyond the scope of the current report and could form the basis of a separate paper. 3/ The number of times experiments were performed are listed.

Minor issues:

1/ In the abstract, perhaps the authors would consider replacing the word "iconic" with "widely recognized".

Authors: we understand where 'iconic' is imprecise. We replaced this word by 'representative', which best states the fact that DnaC and DnaI are at the heart of the replicative helicase loading systems known to date.

2/ The authors mention in their letter that the experiments reported in Figure 3 were performed at least 5 times, but the statement in the paper states that the number of repetition was lower (repeated 3 times with the exception of one experiment that was performed twice). Both statement are acceptable, but is the statement in Figure 3 mistaken?

Authors: The experiments were performed at least 3 times.

Reviewer #3 (Remarks to the Author):

All critical comments have been addressed satisfactorily.